# Duodenal Dual-Wavelength Photobiomodulation Improves Hyperglycemia and Hepatic Parameters with Alteration of Gut Microbiome in Type 2 Diabetes Animal Model

**DOI:** 10.3390/cells11213490

**Published:** 2022-11-03

**Authors:** Se Hee Min, Jinhee Kwon, Eun-Ju Do, So Hee Kim, Eun Sil Kim, Jin-Yong Jeong, Sang Mun Bae, Sang-Yeob Kim, Do Hyun Park

**Affiliations:** 1Division of Endocrinology and Metabolism, Department of Internal Medicine, University of Ulsan College of Medicine, Asan Medical Center, Seoul 05505, Korea; 2Digestive Diseases Research Center, Department of Internal Medicine, University of Ulsan College of Medicine, Asan Medical Center, Seoul 05505, Korea; 3Department of Medical Science, Asan Medical Institute of Convergence Science and Technology, University of Ulsan College of Medicine, Asan Medical Center, Seoul 05505, Korea; 4Convergence Medicine Research Center, Asan Institute for Life Sciences, Asan Medical Center, Seoul 05505, Korea; 5Department of Convergence Medicine, Asan Institute for Life Sciences, University of Ulsan College of Medicine, Asan Medical Center, Seoul 05505, Korea

**Keywords:** type 2 diabetes, duodenum, dual-wavelength, photobiomodulation, light-emitting diode, microbiome

## Abstract

Background: Recently, the duodenum has garnered interest for its role in treating metabolic diseases, including type 2 diabetes (T2DM). Multiple sessions of external photobiomodulation (PBM) in previous animal studies suggested it resulted in improved hyperglycemia, glucose intolerance, and insulin resistance with a multifactorial mechanism of action, despite the target organ of PBM not being clearly proven. This study aimed to determine whether a single session of a duodenal light-emitting diode (LED) PBM may impact the T2DM treatment in an animal model. Methods: Goto–Kakizaki rats as T2DM models were subjected to PBM through duodenal lumen irradiation, sham procedure, or control in 1-week pilot (630 nm, 850 nm, or 630/850 nm) and 4-week follow-up (630 nm or 630/850 nm) studies. Oral glucose tolerance tests; serum glucagon-like peptide 1 (GLP-1), glucose-dependent insulinotropic polypeptide, and insulin levels; liver chemistry and histology; and gut microbiome in the PBM, sham control, and control groups were evaluated. Results: In the 1-week study, duodenal dual-wavelength (D, 630/850 nm) LED PBM showed improved glucose intolerance, alkaline phosphatase and cholesterol levels, and weight gain than other groups. The D-LED PBM group in the 4-week study also showed improved hyperglycemia and liver enzyme levels, with relatively preserved pancreatic islets and increased serum insulin and GLP-1 levels. Five genera (*Bacteroides*, *Escherichia*, *Parabacteroides*, *Allobaculum*, and *Faecalibaculum*) were significantly enriched 1 week after the D-LED PBM. *Bacteroides acidifaciens* significantly increased, while Lachnospiraceae significantly decreased after 1 week. Conclusion: A single session of D-LED PBM improved hyperglycemia and hepatic parameters through the change of serum insulin, insulin resistance, insulin expression in the pancreatic β-cells, and gut microbiome in T2DM animal models.

## 1. Introduction

Type 2 diabetes (T2DM) is a chronic, progressive metabolic disorder characterized by deteriorative hyperglycemia following impaired insulin secretion and utilization. Even though currently available anti-hyperglycemic agents allow glycemic control to some extent, a significant number of patients with T2DM still do not achieve the target glycemic control [1,2]. Notably, none of the currently available therapeutic agents can reverse the pathophysiology of T2DM. Roux-en-Y gastric bypass (RYGB), the standard surgical procedure for bariatric surgery, is characterized by restriction of stomach volume, bypass of the entire duodenum and proximal jejunum, and rapid movement of a large volume of unabsorbed nutrients to the distal small intestine [3]. Due to the favorable effects of bariatric surgery, the recent consensus statements on T2DM and morbid obesity advocated bariatric surgery as a treatment option for patients for whom medical treatments do not adequately control their disease [4]. However, only a small proportion of patients with T2DM undergo bariatric surgery due to the invasiveness and irreversibility of the procedure [5]. Therefore, exploiting effective but less invasive procedures that could produce similar metabolic effects to bariatric surgery could be advantageous in addressing unmet clinical needs.

External photobiomodulation (PBM) with light-emitting diode (LED) in rat or mouse models may result in improved glucose tolerance, insulin resistance and fasting hyperinsulinemia, and regeneration of the pancreas and liver in previous animal models [6,7,8,9,10,11,12]. External PBM also reduced adiposity and inflammatory infiltrate in adipose tissue [7,13,14]. This effect may be associated with increased adenosine triphosphate in mitochondria and biological response in cells by PBM [15]; however, it is not clearly proven [16,17]. The targeted organ in external PBM in rat or mouse models is also unclear due to the nonspecific external irradiation of LED in the rat model. Although a biphasic dose response in external PBM with low energy density and multiple sessions may be beneficial [10,14], external PBM may be required for multiple sessions with a relatively large power density (75–100 mW/cm^2^) and may be unsuitable for large animal studies or human trials with large body surfaces compared to rat or mice models [18,19,20].

Recently, the duodenum has been gaining attention as the target organ for endoscopic bariatric and metabolic therapies, including duodenal bypass surgery or hydrothermal radiofrequency ablation (RFA) for duodenal mucosal resurfacing (DMR) [21,22,23]. Particularly, DMR may be considered an adjunctive nonpharmacologic treatment for T2DM and non-alcoholic steatosis due to its durable outcomes. The findings of a prospective, multicenter trial with patients with T2DM showed that endoscopic DMR decreased the levels of hemoglobin A1c, fasting plasma glucose, homeostasis model assessment-insulin resistance (HOMA-IR), body weight, and hepatic enzymes [22]. The effects of metabolic improvement were sustained even after 12 months [22]. However, repeated DMR may raise concern because repeated RFA in duodenal mucosa may have a higher risk of stricture formation and procedure-related adverse events, and the procedure seems highly operator-dependent, thus related to substantial rates of technical failure [21,22]. Therefore, a non-thermal duodenal approach as a nonpharmacologic treatment for durable glycemic control may be an interesting alternative for hydrothermal DMR in terms of potentially repeated and safe endoscopic procedures. We conducted this study in an animal model of T2DM to determine whether duodenal LED PBM affects glycemic control and hepatic parameters.

## 2. Materials and Methods

### 2.1. Animals

Diabetic Goto–Kakizaki (GK) rats (8–9 weeks old, male) were purchased from Japan SLC Inc. (Hamamatsu, Japan). We employed GK rats, a non-obese type 2 diabetes animal model, that spontaneously develop hyperglycemia due to progressive pancreatic β–cell failure and increased peripheral insulin resistance [24,25]. The animals were randomly selected and housed in individual cages to monitor their food intake under constant ambient temperature and humidity in a 12-h light/dark cycle. More details about the experimental protocol and LED-PBM device specifications are described in the Appendix A.

For the energy setting of duodenal PBM, the dose clinically used for oral mucosa or skin (2–5 J/cm^2^) was adopted for duodenal mucosa [26] with less than 10 min of irradiation time as the acceptable procedure time of upper endoscopy for application [27] in future large animal studies and human trials (Appendix A). The light power was measured with a calibrated detector (Newport 918D-ST-SL wand detector from 400 to 1100 nm; Newport Corp., Irvine, CA, USA) and power meter (Newport 1936-R; Newport Corp., CA, USA). Based on previous reports of PBM regarding improved glucose intolerance [7,8,9,10] and the availability of LED chips for application to a catheter in our laboratory, we used an LED with wavelengths of 630 nm and 850 nm.

In the 1-week follow-up study, an oral glucose tolerance test (OGTT) was conducted at baseline and 1 week after in 3 groups based on the procedure performed: a single session of duodenal PBM with a two-dimensional illuminated LED in a triangular catheter (5 cm length, 630 nm [n = 8]: irradiance, 3.70 mW/cm^2^, energy density, 2.22 J/cm^2^, and 600 s; 850 nm [n = 8]: irradiance, 32.72 mW/cm^2^, energy density, 3.60 J/cm^2^, and 100 s; 630/850 nm [n = 10]: 1.85 J/cm^2^, 3.85 J/cm^2^, and 500 s) (Appendix A), RFA (n = 5, catheter-based, 2.2 cm in exposure length, 80 °C, 7 W, 5 s; Taewoong Medical, Seoul, Korea), and sham control (n = 5, laparotomy with gastrostomy and placement of a non-illuminated catheter with the same time as that of duodenal LED PBM group) groups. For duodenal LED PBM, a two-dimensional illuminated LED with 5 cm in exposure length in the proximal duodenum was manually placed toward the pancreas, which was located on the ipsilateral and medial side of the duodenum. During the procedure, to identify the thermal effect of LED and RFA, body temperature was monitored using an infrared thermometer (FLIR E6-XT; FLIR Systems Inc., Wilsonville, OR, USA). The thermometer was set to indicate the highest temperature at the point of interest.

One week after OGTT, a necropsy was performed to examine the entire duodenum, liver, and pancreas. For the measurements of the long-term effects of glycemic control by duodenal LED PBM, we measured metabolic profiles after a 4-week follow-up study. In the 4-week follow-up study, a single session of duodenal PBM with four-dimensional illuminated LEDs with 5 cm in exposure length (630 nm: n = 5, 630/850 nm: n = 6, sham control: n = 5, control: n = 5) was performed to uniformly irradiate the whole surface of proximal duodenum. (Appendix A) In brief, in the experiment of 1 week, the triangle-type catheter with a two-dimensional encapsulated LED surface could not cover the duodenum evenly with light. So we added two more encapsulated LED surfaces as four dimensions close to a polyhedron to irradiate from all directions to cover the entire area in the experiment of 4 weeks. For this hypothetical background, double the energy density in the 4-week experiment compared to the 1-week experiment regarding single or dual wavelength was applied.

Based on the results of the 1-week pilot study, the optimal wavelength of LED PBM was selected for the 4-week follow-up study (Appendix A). OGTT, tests to measure serum glucagon-like peptide 1 (GLP-1), glucose-dependent insulinotropic polypeptide (GIP), and insulin levels, and liver chemistry were performed at baseline, 1, and 4 weeks after duodenal PBM in the 4-week follow-up study, control, and sham control groups. Necropsy was performed to evaluate these groups’ duodenal mucosa, liver, and pancreas after 4 weeks.

### 2.2. Oral Glucose Tolerance Test (OGTT)

After fasting overnight (16–18 h), an OGTT was performed twice in the pilot study (at 0 and 1 week after PBM) and at three time points (at 0, 1, and 4 weeks after PBM) in the 4-week follow-up study. Blood was collected from the tail vein. Following baseline, a solution of 25% dextrose (0.01 mL/g) was orally administered at 2.5 g/kg. Blood glucose levels were measured at baseline, 15, 30, 60, 90, and 120 min, with or without 180 min.

### 2.3. Biochemical Analysis

All biochemical and hormonal parameters were measured in collected plasma samples from the OGTT. The chemical analysis was carried out to determine the concentration of liver enzymes such as cholesterol, aspartate aminotransferase (AST), alanine aminotransferase (ALT), and alkaline phosphatase (ALP) using 7189 Clinical Analyzer (HITACHI, Tokyo, Japan). The hormonal parameters, including insulin, GLP-1, and GIP, were estimated using the following enzyme-linked immunoassay (ELISA) kits, respectively: Rat Insulin ELISA Kit (CC-90010, Crystal chem, Elk Grove Village, IL, USA), Multi Species GLP-1 Total ELISA (EZGLP1T-36K, Merck Millipore, Burlington, MA, USA), and Rat/Mouse GIP (total) ELISA (EZRMGIP-55K, Merck Millipore, Burlington, MA, USA). Insulin, GLP-1, and GIP levels were calculated following the manufacturer’s instructions. Each assay was performed in duplicate from the indicated samples.

### 2.4. Histology

Histologic examination of the duodenum, liver, and pancreas was performed after the necropsy. The excised tissues were fixed in 10% neutral buffered formalin. Fixed tissues were rinsed with tap water to remove the formalin for about 2 h. The tissues were then dehydrated in graded ethanol, cleared in xylene using a tissue processor (Excelsior AS Shandon Diagnostics Ltd., Runcorn, UK), and embedded into paraffin blocks using a paraffin embedding station (Leica, Wetzlar, Germany, EG1150H). Paraffin blocks were cut into 3 µm thick sections using a rotary microtome (Leica, RM2255). Sections were prepared onto the glass slides. Hematoxylin and eosin (H&E) staining, Masson trichrome (MT) staining, and multiplex immunohistochemistry (IHC) staining were performed (detailed protocol is uploaded in Appendix A). H&E- and MT-stained sections were digitized using a slide scanner (VS200; Olympus, Tokyo, Japan). The MT-stained section of the liver tissue was examined to measure interstitial collagen volume fraction (blue dyed) and the corresponding area occupied (red dyed) using an image analysis software (VS20S-DESK v3.2; Olympus, Tokyo, Japan). The insulin IHC-stained pancreatic section was observed using software (VS20S-DESK v3.2; Olympus, Tokyo, Japan) to estimate the number and size of stained islet cells.

### 2.5. Microbiome Analysis

Stool samples for microbiome analysis were collected from rats at baseline and 2 days, 8 days (1 W), 16 days, and 27-days (4 W) after the procedure. The stool was stored at –80 °C until analysis. Total genomic DNA from stool samples was extracted by the manufacturer’s instructions [28]. The V4 region of the 16S rRNA gene was amplified using 515 F and 806 R primers designed for dual indexing and pooled, and the Illumina iSeq100 platform was used for the analysis. Processing raw sequencing reads started with a quality check, and filtration of low-quality (<Q25) reads using Trimmomatic ver. 0.32. Nonspecific amplicons that do not encode 16S rRNA were detected by nhmmer in the HMMER software package ver. 3.2.1 with hmm profiles. Unique reads were extracted, and redundant reads were clustered with the unique reads by the derep_fulllength command of VSEARCH. Chimeric reads were filtered on reads with <97% similarity by reference-based chimeric detection using the UCHIME algorithm and the non-chimeric 16S rRNA database from EzBioCloud. Cluster_fast command was used to perform de-novo clustering to generate additional operational taxonomic units. The secondary analysis, which includes diversity calculation and biomarker discovery, was conducted with in-house programs from CJ Bioscience, Inc. (Seoul, South Korea).

### 2.6. Statistical Analysis

Continuous variables are expressed as means with interquartile ranges (IQR). All sample size in each group was set over 5 rats to conduct significant statistics. An unpaired *t*-test and an ordinary one-way ANOVA test were used to examine the difference between the PBM group and other groups. In each group, the Wilcoxon and Kruskal–Wallis tests were conducted to evaluate the differences between 1- and 4-week follow-ups. A two-sided *p*-value of <0.05 indicated statistically significant differences for all analyses. To evaluate the greatest differences in bacterial taxa between the groups including control, sham control, and LED PBM group, linear discriminant analysis (LDA) effect size (LEfSe) was performed that a Wilcoxon rank sum test combined with linear discriminant analysis [29]. The LEfSe analysis determined significant ranking (*p* < 0.05) of abundant microbiomes where a log (10) transformed effect size was greater than 2. Statistical analyses were performed using GraphPad Prism 9.0.0 software (GraphPad Software Inc., San Diego, CA, USA).

## 3. Results

### 3.1. Baseline Characteristics of Duodenal PBM and Other Treatment Groups

Median procedure time, including surgical incision, catheter intubation in the duodenum, and closure for duodenal PBM or sham control, was 32.4 min (IQR: 29.1–35.0) in the 1-week pilot study and 29.6 min (IQR: 25.0–33.3) in the 4-week follow-up study. All rats included in the case group treated with duodenal LED PBM and sham control survived during and after the procedure without any complications and signs of sepsis, peritonitis, or wound infection. Four different energy settings were performed in the RFA group of the pilot study, with different irradiated times of 30, 20, 10, and 5 s, respectively (7 W, 30 s; n = 2, 20 s; n = 3, 10 s; n = 4, 5 s; n = 5). Only five rats that were irradiated for 5 s during the time-variable energy setting survived, with mild intestinal stenosis showing on necropsy 1 week following duodenal RFA. The other rats (n = 9: 30 s; n = 2, 20 s; n = 3, 10 s; n = 4) irradiated with the RF probe died due to several intestinal obstructions on the day of the implementation or the next day. No significant body temperature change was seen in the LED PBM group compared to that in the RFA group (Appendix A).

### 3.2. Effects on Metabolic Parameters in the Pilot Study

Figure 1 shows the pilot study results related to the OGTT, body weight change, biochemistry, and histological analysis. Duodenal dual-wavelength (D, 630/850 nm)-LED PBM treated rats showed better glucose tolerance compared to all other study groups, as shown by −9.7% lower area under the curve (AUC) in glucose levels on the OGTT curve (Figure 1A, *p* < 0.01) and lower 15- and 30-min time points as compared to sham control-treated rats (Figure 1B, *p* < 0.005 for both). No significant differences in AUC in glucose on the OGTT curve in the sham control, RFA, 630 nm, and 850 nm groups after 1 week were observed (Figure 1B), even though the duodenal LED PBM with 630 nm and RFA groups showed lower glucose levels at 15 (*p* < 0.005), 30 (*p* < 0.05), and 120 min (*p* < 0.005) (Figure 1A). In histologic analysis, the collagen deposition presenting a blue area in the liver tissue excised from the D-LED PBM group was investigated at a lower value (0.01%) than that from the sham control (0.16%) and RFA groups (0.24%) without statistical significance (Figure 1C). In comparison to the sham control group, only the D-LED PBM group showed a significantly lower body weight gain (Figure 1D, *p* < 0.05) without a change in food intake (Appendix A). Sham control and PBM groups showed no significant difference in AST and ALT levels compared with those from baseline to 1 week after PBM. The cholesterol and ALP levels 1 week after PBM were significantly less than those at baseline in the D-LED PBM and sham control groups (cholesterol: *p* < 0.05, ALP: *p* < 0.01) (Figure 1E).

### 3.3. Effects on Metabolic Parameters in the 4-Week Follow-Up Study

Based on the results of the 1-week pilot study, 630 nm and 630/850 nm were selected for duodenal LED PBM in the 4-week follow-up study. After 1 week of treatment, no significant differences in glucose levels in the OGTT curve were observed compared to baseline levels in the control, sham control, and 630 nm groups (Figure 2A). However, the D-LED (630/850 nm) PBM group showed a significantly lower glucose level at 15- (*p* < 0.05) and 30-min (*p* < 0.05) time points after 1 week, and the treatment effects were maintained at 15- (*p* < 0.05), 30- (*p* < 0.005), and 60-min (*p* < 0.01) time points for 4 weeks (Figure 2A). Accordingly, the areas under the serum glucose concentration–time curve (AUC glucose) of the D-LED PBM group at both periods were significantly lower than those at baseline, 11.68% (*p* < 0.01) lower after 1 week, and 14.50% (*p* < 0.005) lower after 4 weeks (Figure 2B). Even though no difference in the percentage of body weight gain among all the treatment groups was observed (Figure 2C), the D-LED PBM group showed significantly reduced levels of liver enzymes such as AST (*p* < 0.05) and ALT (*p* < 0.01) at 4 weeks post-treatment compared to those at baseline, not observed in the control and sham control groups (Figure 2D).

### 3.4. Effects on the Pancreatic Islets, Liver Fibrosis, and Duodenum in the 4-Week Follow-Up Study

The progressive loss of β cells in the pancreatic islets with fibrosis was clearly observed in the control group (Figure 3A). However, the pancreatic islets were relatively well preserved, and expression of insulin in IHC was increased 4 weeks after treatment in the D-LED PBM group (Figure 3A). The percentage of β-cell areas within each pancreatic section was significantly greater in the D-LED PBM group (*p* < 0.01) than that in the control group (Figure 3B). HOMA-IR in the D-LED PBM group significantly decreased at 1 (*p* < 0.005) and 4 weeks (*p* < 0.01) after treatment compared to the control and sham control group (Figure 3C). There was no significant difference in collagen deposition in the liver among all treatment groups. No definite mucosal abnormality or duodenal wall injury was seen in the D-LED PBM group. No statistically significant difference was observed in the expression of GIP and GLP-1 in duodenal villi among the control, sham control, and D-LED PBM groups on IHC staining (Figure 4).

### 3.5. Composition of the Gut Microbiome in the 4-Week Follow-Up Study

As a result of the alpha-diversity analysis of the control group, the diversity of bacteria in the control species increased after 1 week. However, after 4 weeks, the trend was similar to that of the baseline. Similar to the control group, in the sham control group, the species diversity, according to the results of the alpha-diversity analysis, increased after 1 week compared to that before baseline, and the species diversity after 4 weeks showed a pattern similar to that after 1 week. However, the D-LED PBM group showed different alpha-diversity analysis results from the control or sham control groups. In the D-LED PBM group, the species diversity decreased after 1 week compared to that before duodenal LED PBM. After 4 weeks, the species diversity slightly increased without significant differences among the groups; however, the species diversity decreased compared to that before the duodenal LED PBM (Figure 5A). In the beta-diversity analysis using the Unifrac distance metric in the three groups (control, sham control, D-LED PBM group), the gut microbiota was not clearly distinguished based on the time points (Figure 5B).

Next, the percentage of bacterial taxa discriminating the gut microbiota was calculated according to the time period after the duodenal LED PBM metric in the three groups (control, sham control, and D-LED PBM group). The gut microbiota of the control group was abundantly distributed in the order of Firmicute, Bacteroidetes, Proteobacteria, and Actinobacteria at the phylum level. In the control group, after 1 week of PBM, Firmicutes and Proteobacteria decreased, and Bacteroidetes tended to increase. The composition ratio of microbiota at the phylum level in the sham control group was similar to that of the control group. In the D-LED PBM group, after 1 week, Firmicutes decreased, and Bacteroidetes and Proteobacteria increased. However, in the D-LED PBM group, no significant change in gut microbiota composition at the phylum level was observed over time (Figure 5C). At the family level, the most abundant gut microbiome in the three groups was Erysipelotrichaceae, which belongs to Firmicutes, followed by Murbaculaceae, which belongs to Bacteroidetes (Figure 5D). Through linear discriminant analysis Effect Size, significant gut microbiota at the genus level before and 1 week after D-LED PBM was presented as a taxonomic bar chart (Figure 5E–G). Five genera (*Bacteroides*, *Escherichia*, *Parabacteroides*, *Allobaculum*, *Faecalibaculum*) were significantly enriched after 1 week in the D-LED PBM group (*p* < 0.05). In the control group, three genera (KE1591_g, PAC001516_g, PAC001165_g) were significantly enriched after 1 week (Figure 5E, *p* < 0.05). In the sham control group, six genera (*Bacteroides*, *Pseudoflavonifractor*, *Eubacterium*_g23, *Parabacteroides*, *Escherichia*, and *Alistipes*) were significantly enriched after 1 week (Figure 5F, *p* < 0.05). On the other hand, in the D-LED PBM group, five genera belonging to Lachnospiraceae decreased significantly after 1 week (Figure 5G, *p* < 0.05). Characteristically, in the D-LED PBM group, Bacteroides acidifaciens was significantly increased after 1 week (Appendix A, *p* < 0.05).

### 3.6. Association of Altered Gut Microbiome and GIP, GLP-1, Insulin Resistance in Weeks

Dynamic changes in serum GIP, GLP-1, insulin, and insulin resistance between 1 and 4 weeks after PBM, measured using HOMA-IR, were identified for the evaluation of underlying mechanisms on altered gut microbiome over 1–4 weeks in the D-LED PBM group [30]. Although the differences in serum GIP, GLP-1, and insulin levels during the OGTT were not significant between baseline and the 4-week post-treatment periods, the differences in the AUC insulin (*p* < 0.01) between baseline and 1 week and between 1 week and 4 weeks of treatment, and in the AUC GLP-1 (*p* < 0.01) between the 1 and 4 weeks of treatment were significantly higher in the D-LED PBM groups compared with that in the control and sham control groups (Figure 6A,B). Additionally, in the D-LED PBM group, HOMA-IR, as the parameter of insulin resistance, significantly decreased at 1 (*p* < 0.005) and 4 weeks (*p* < 0.01) after treatment compared with the baseline (Figure 3C). The AUCs of serum GIP levels during OGTT did not differ before and after each treatment (Figure 6C).

## 4. Discussion

This study, using the rodent model of T2DM, aimed to evaluate the role of duodenal LED PBM for non-pharmacological treatment. In this study, D-LED PBM showed improved serum glucose levels and hepatic parameters, maintained for 4 weeks after a single session of D-LED PBM. GK rats share a number of similarities with human type 2 diabetes, as they are characterized by both insulin resistance and impaired insulin secretory function [24]. They spontaneously develop early glucose intolerance accompanied by impaired glucose-stimulated insulin secretion and increased hepatic glucose production during the postweaning period. Muscle and adipose tissue insulin resistance also develops after 2 months of age [25]. Similar to human type 2 diabetes, GK rat islets are characterized by increased progressive β–cell destruction and fibrosis [24,25]. Thus, based on the results of this study, duodenal LED PBM could be potentially applied for treating patients with type 2 diabetes.

LED PBM could be used as safe and effective phototherapy for patients with diabetic foot [11,30]. Previous animal studies showed that external LED PBM to the pancreas in streptozotocin-induced diabetic rodent models increased the cell density of islets and pancreatic ducts, stimulated hepatic glycogenesis, and modified carbohydrate metabolism [11,14]. In a previous ex vivo study, external irradiation of infrared LED affected the secretion of insulin in the pancreas despite not being pancreas-targeted internal irradiation, possibly weakening treatment effectiveness [7].

No ideal wavelength, including dosimetry or single-point vs. full-body irradiation with LED for external PBM to treat insulin resistance, has been established [31]. Approximately 600 nm or 800 nm is usually applied for external PBM in animal models [7,8,9,10,14]. Combination of 600 and 800 nm LED PBM has also been studied owing to different cell signaling and tissue penetrating ability [8,15]. In this pilot and 4-week follow-up study, D-LED PBM showed a consistent reduction in serum glucose levels after duodenal LED PBM. Compared to a single session of duodenal single-wavelength LED (630 nm or 850 nm) PBM, D-LED PBM showed a significant reduction in serum glucose levels, approximately 10% and 15%, in the AUC curve of OGTT in the 1-week pilot and 4-week follow-up studies, respectively. The combination of 630 nm and 850 nm LED PBM may have a different role in the change of gut microbiome and insulin resistance after PBM by 630 nm, and an increased expression of insulin in the pancreas islet cell and serum insulin level from 1 to 4 weeks after duodenal LED PBM by 850 nm with deep penetrating ability from the duodenum to pancreas parenchyma may cause improved serum glucose and hepatic parameters [7,9,10,23,32,33]. This potential pluripotent role of D-LED PBM compared to single-wavelength duodenal LED PBM needs further evaluation. Except for the dual wavelength, the groups did not show statistical differences in the AUC of OGTT between baseline and 1 week even though 850 nm (−5.33%) showed an approximately 4 times greater reduction than 630 nm single group (−1.20%) with a large standard deviation (630 nm: 14.4, 850 nm: 9.5) compared to the 4-week follow-up group. Based on the 1-week experiment, we decided to determine the optimal wavelength (single or dual) for duodenal PBM in the 4 weeks experiment. Despite the results for the single (630 or 850 nm) wavelength, the duodenal PBM was not statistically significant for the AUC of OGTT, but the 630 nm single group showed significant differences from the OGTT curve at 15 min and 30 min, whereas the 850 nm single group showed no significant difference. (Figure 1A). This is the reason why we chose a single 630 nm rather than 850 mm isolated for the 4 weeks experiment.

Our results revealed that the AUC in blood glucose levels during the OGTT markedly decreased after 4 weeks of one session of D-LED PBM compared to the baseline OGTT curve before D-LED PBM. Interestingly, the serum insulin levels during OGTT increased, while HOMA-IR decreased after D-LED PBM. These results might be explained by enhanced insulin secretion with reduced insulin resistance after D-LED PBM. Accordingly, pancreatic islets in the D-LED PBM group were better preserved than those in the control group. It is possible that D-LED PBM contributes to β cell regeneration and insulin secretion by stimulating duodenal enteroendocrine cells and increasing endogenous gut hormones that regulate glucose homeostasis. The serum level of GLP-1 and insulin were increased at 4 weeks compared to those at 1 week in the D-LED PBM group, indicating that D-LED PBM improves the potentiation of glucose-mediated insulin secretion by GLP-1. However, there was no difference in serum levels of GIP after treatment. Whether D-LED PBM acts directly on proliferating β cells from the pancreas or indirectly stimulates the pancreas by modulating other gut hormone levels is unclear and needs further examination to identify the precise mechanism. In addition, D-LED PBM might contribute to reduced insulin resistance by sharing mechanisms suggested for metabolic improvements after RYGB, including reduction of hepatic glucose production, alterations of bile acid metabolism, attenuated metabolic endotoxemia via decreased intestinal permeability, and altered host-microbial interactions [34,35,36]. Given the long-lasting effect on lowering serum glucose levels for 4 weeks following a single session of D-LED PBM with lower power (3.7–7.7 mW/cm^2^) compared to multiple sessions of extra-body LED PBM with relatively larger power (75–100 mW/cm^2^) in previous animal studies [7,9,10,15,32], the duodenum may be a significant therapeutic target for PBM for metabolic diseases including T2DM.

Interestingly, the 4-week follow-up study group with four-dimensional LED had two-fold total energy compared to the 1-week pilot study group. However, the reduction of hyperglycemia during OGTT in the 4-week follow-up study was modestly increased compared to that in the 1-week pilot study after 1 week of D-LED PBM (−9.7% in the 1-week pilot study vs. −11.7% in the 4-week follow-up study) without a significant difference (*p* value = 0.57). Therefore, it is not clear that uniform irradiation with double total energy in D-LED PBM may proportionally impact the degree of reduction of serum glucose levels. We designed this study by separating it into two observation periods, a 1-week and 4-week follow-up. The pilot study was implemented to determine the safety and feasibility of the LED catheter application and the appropriateness of the treatment effect. Within the short-term period, pathological analysis was required to confirm whether there was pathological damage at the treatment site. In addition, the reason we inserted a squared LED catheter was to improve the effectiveness of irradiance. Because this pilot study focused on the feasibility of duodenal LED PBM with a single or dual wavelength, further experiments appropriately designed for the energy density to determine if a dual wavelength is better than a single wavelength for duodenal LED PBM may be required.

In the comparison of the several previous studies about external irradiance for PBM [7,10,13], we targeted the descending duodenum, which is partially located at the retroperitoneum as a tubular organ, but we found it difficult to irradiate the duodenum uniformly and efficiently to provide external delivery of PBM in future large animal studies and human trials. This is a proof-of-concept study for designing future large animal studies and human trials for duodenal LED PBM. Therefore, there is practically no justification for illumination through a light guide in this study. Further external and internal PBM preclinical studies with various wavelengths with energy densities and irradiance for targeting the duodenum, liver, gut microbiome, and pancreas would be of interest.

One of the novel findings of this study is the evidence for alteration of the gut microbiome after D-LED PBM. Five genera (*Bacteroides*, *Escherichia*, *Parabacteroides*, *Allobaculum*, and *Faecalibaculum*) were significantly enriched after 1 week in the D-LED PBM group. Particularly, *Bacteroides acidifaciens* significantly increased after 1 week. On the other hand, five genera belonging to the Lachnospiraceae family significantly decreased after 1 week. In a previous report [32], infrared-wavelength PBM increased *Allobaculum*, a bacterium associated with a healthy microbiome, as in our study. Given no enhanced expression of GIP and GLP-1 in duodenal mucosa after 4 weeks following D-LED PBM, alteration of the gut microbiome, dynamic changes in serum GLP-1 and insulin levels, and improved insulin resistance as shown by HOMA-IR from 1 week to 4 weeks in the D-LED PBM group show that duodenal LED PBM may directly affect alterations of the gut microbiome, resulting in lowering serum glucose levels, rather than affecting the duodenal mucosa. Furthermore, the reduction in the glucose AUC (−11.7%) in OGTT in 1 week was relatively lower than that at 4 weeks (−14.5%) in the D-LED PBM group of the 4-week follow-up study. Based on alterations of the microbiome between baseline and 1-week as the early phase of dynamic change in the gut microbiome, insulin resistance was improved after 1 week of treatment as reflected by the HOMA-IR (Figure 3C), and this may affect the normalized AUC percentage of insulin that decreased significantly and improved the hyperglycemia from baseline to 1 week in the D-LED PBM group. In the late phase, within 4 weeks, improved insulin resistance was maintained after 4 weeks of D-LED PBM based on HOMA-IR (Figure 3C) and increased expression of insulin in IHC of the pancreas after 4 weeks of D-LED PBM (Figure 3B) as multifactorial effects may potentiate reductions in the serum glucose level. Further studies on the association between the dynamic alterations of the gut microbiome and insulin resistance in D-LED PBM may be required.

Bacteroides *acidifaciens* have been reported to be associated with weight and fat loss, increased insulin, and decreased glucose [37]. Additionally, when RYGB, which is one of the effective treatments for morbid obesity and T2DM, was performed, Bacteroides acidifaciens tended to increase, as shown in this study [38]. Bacteroides acidifaciens may have a direct effect on serum glucose changes. After D-LED PBM, Allobaculum and Bacteroides acidifaciens were increased, as described above, while bacteria belonging to Lachnospiraceae family were significantly decreased. According to Kang et al. [39], Lachnospiraceae_UCG_005 was significantly abundant in the stomach of GK rats compared with that in non-diabetic Wistar rats. Furthermore, blood glucose levels were negatively correlated with Lachnospiraceae_UCG_005. These suggest that the changes in gut microbial composition after D-LED PBM are closely related to the lowering of serum glucose levels and the improvement of hyperglycemia. Therefore, a further study may be needed to determine whether probiotics or postbiotics that mimic the microbial effect of D-LED PBM maintained improved hyperglycemia, insulin resistance, and hepatic parameters after D-LED PBM.

T2DM and non-alcoholic fatty liver disease (NAFLD) are known to be closely linked through insulin resistance as a common pathological driver [40]. These two metabolic conditions often co-exist in patients. Since we used the GK rat, which is a non-obese model of T2DM, we could not observe any changes in body weight or liver fat content in the 4-week follow-up study. However, the liver enzymes such as AST and ALT were reduced in DM-LED PBM (Figure 2D), which might be indirect effects of metabolic improvements associated with the attenuation of insulin resistance and glucose toxicity by DM-LED PBM. Further investigations using animal models of diet-induced obesity or NAFLD are needed to elucidate the direct effects of LED-PBM on the liver.

Although this study highlights a very interesting target in the body to be irradiated to treat diabetes, adjustments in light parameters to irradiate this target tissue through the skin seems to be the best option, mainly due to the need for repetitive sessions of photobiomodulation as a noninvasive treatment. However, this pilot study with laparotomy-guided duodenal LED PBM was designed for the feasibility of D-LED PBM and future applications in large animal and human clinical trials with endoscopic duodenal LED PBM. This endoscopic duodenal LED PBM may be an attractive platform in a country that has widely available routine upper endoscopy for cancer screening in terms of repetitive sessions and easy access. To bring endoscopic PBM to clinical applications for treating patients with T2DM, further considerations with respect to its efficacy and safety may be required. To improve endoscopic accessibility of the duodenum with LED PBM, the micro LEDs may need to be more flexible to deliver more precise and uniform irradiation to the target organ [41]. In addition, regarding the power of LED for enhancing the biological effects without any harm to the duodenum, such as thermal injury or intestinal stricture, the selection of the optimal-wavelength LED may be needed. Further studies including D-LED PBM using optimal energy from the 4-week follow-up study and transparent cap with micro-LED for even irradiation on the duodenum in large animal studies are needed [41].

## 5. Conclusions

We conclude that a duodenal dual-wavelength light-emitting diode photobiomodulation improves hyperglycemia and hepatic parameters through photobiomodulation-mediated change of serum insulin, insulin resistance, insulin expression in the pancreatic β-cells, and the gut microbiome. Development of endoscopic-assisted duodenal dual-wavelength light-emitting diode photobiomodulation and probiotics or postbiotics that mimic the microbial effect of a D-LED PBM may be encouraged for durable glucose-lowering benefits in patients with T2DM.

## Figures and Tables

**Figure 1 cells-11-03490-f001:**
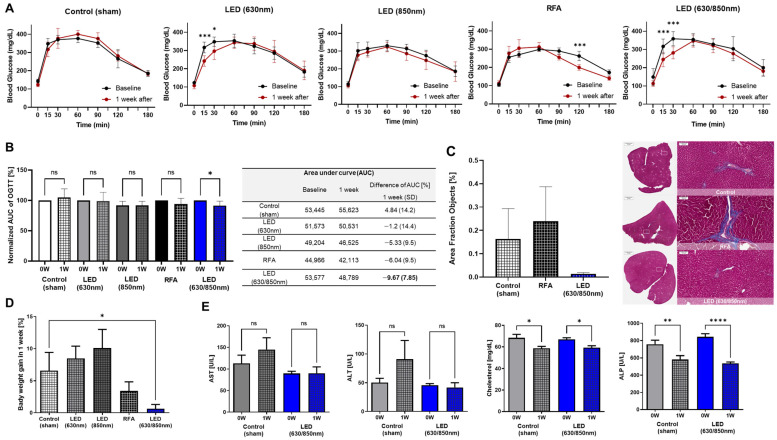
Effects on metabolic improvement in the pilot study with duodenal dual-wavelength (D) photobiomodulation (PBM) with light-emitting diode (LED): Glycemic curves (**A**) during oral glucose tolerance test (OGTT) and normalized area under the curve (**B**) showing the comparison of baseline and 1 week after the procedure in sham control, PBM with 630 nm, PBM with 850 nm, radiofrequency ablation, and PBM with D-LED (630/850 nm) groups. (**C**) In accordance with histological images of Masson’s trichrome staining of the liver tissue, collagen deposition in the liver tissue with D-LED PBM was the lowest compared with those in the sham control and RFA. (**D**) The percentage of body weight gain after the end of the experiment (1-week follow-up). € Comparison of the biochemistry parameters of aspartate aminotransferase, alanine aminotransferase, alkaline phosphatase, and cholesterol levels in sham control and D-PBM groups between 0 and 1 week after PBM. * *p* < 0.05; ** *p* < 0.01; *** *p* < 0.005; **** *p* < 0.001.

**Figure 2 cells-11-03490-f002:**
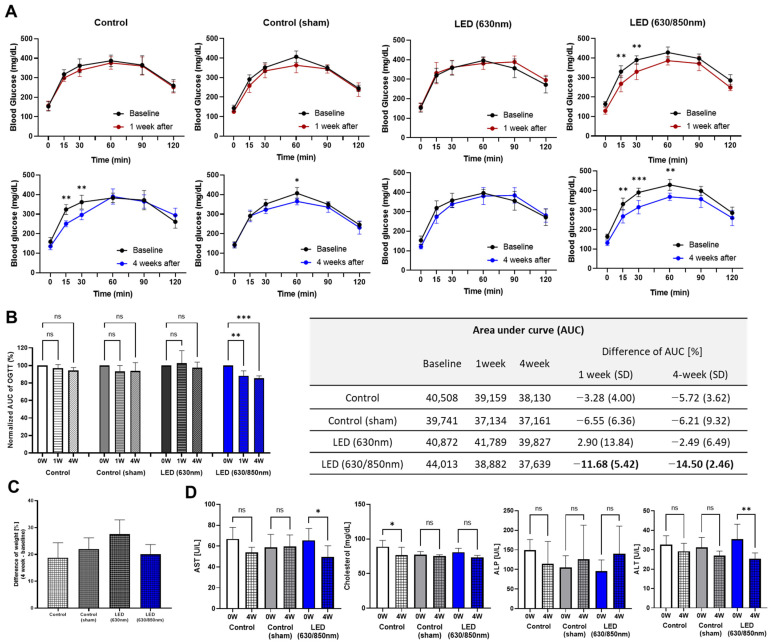
Effects on metabolic improvement in the 4-week follow-up study with duodenal dual-wavelength (D) photobiomodulation (PBM) with light-emitting diode (LED): Glycemic curves (**A**) during oral glucose tolerance test (OGTT) and normalized area under the curve (**B**) showing the comparison of 0 versus 1 week after PBM, and 0 versus 4 weeks after PBM in control, sham control, PBM with 630 nm, and PBM with D-LED (630/850 nm) groups. (**C**) The percentage of body weight gain at the end of the experiment in the 4-week follow-up. (**D**) Comparison of the biochemistry parameters of aspartate aminotransferase, alanine aminotransferase, alkaline phosphatase, and cholesterol levels in control, sham control, and D-PBM groups between 0 and 4 weeks after PBM. * *p* < 0.05; ** *p* < 0.01; *** *p* < 0.005.

**Figure 3 cells-11-03490-f003:**
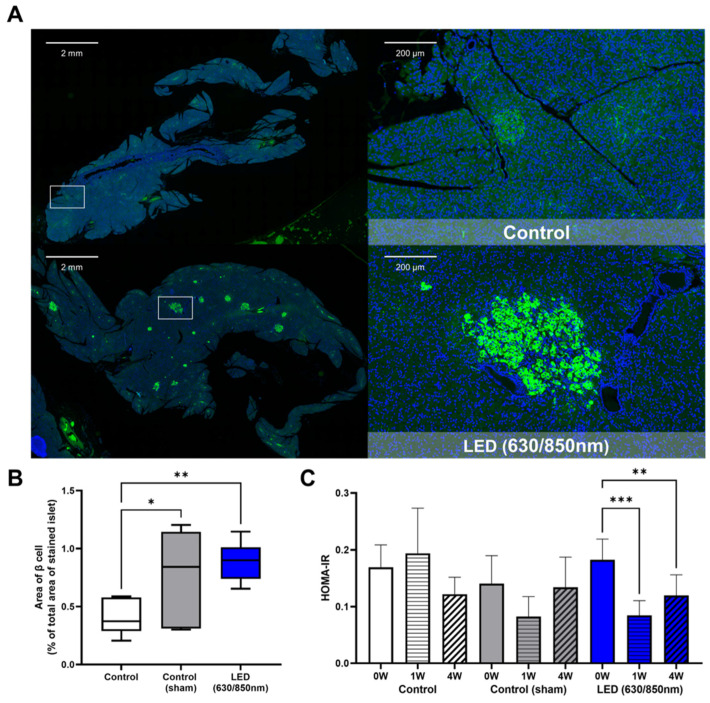
(**A**) Representative pancreatic islets (selected islet in the white box) stained with immunohistochemistry for insulin (β cell) from the control and duodenal dual-wavelength (630/850 nm) PBM (low magnification: 0.4×, high magnification: 5×). (**B**) Comparison of quantitative percentage of the total area of stained pancreatic islets in control, sham control, and D-PBM at 4 weeks after the experiment. (**C**) HOMA-IR of the three groups group at 0, 1, and 4 weeks. Values are expressed as mean ± SEM of five rats in the control and sham control and six rats in the D-PBM group. * *p* < 0.05; ** *p* < 0.01; *** *p* < 0.005.

**Figure 4 cells-11-03490-f004:**
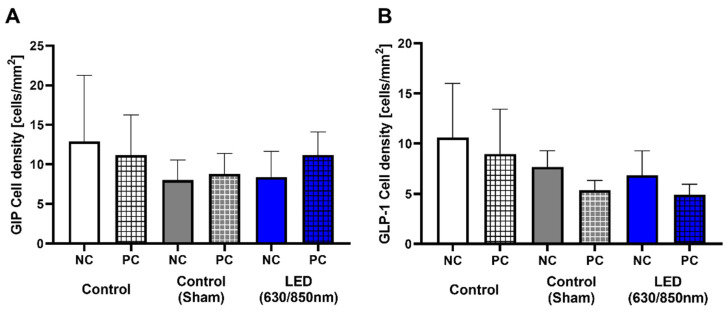
Quantitative measurement of the difference in the expression of (**A**) glucose-dependent insulinotropic polypeptide (GIP) and (**B**) glucagon-like peptide 1 (GLP-1) in duodenal villi, which separately sampled two regions of the duodenum. PC (positive treated tissue) and NC (negative treated tissue) are sampled from duodenum tissue among the control, sham control, and D-PBM (630/850 nm) groups.

**Figure 5 cells-11-03490-f005:**
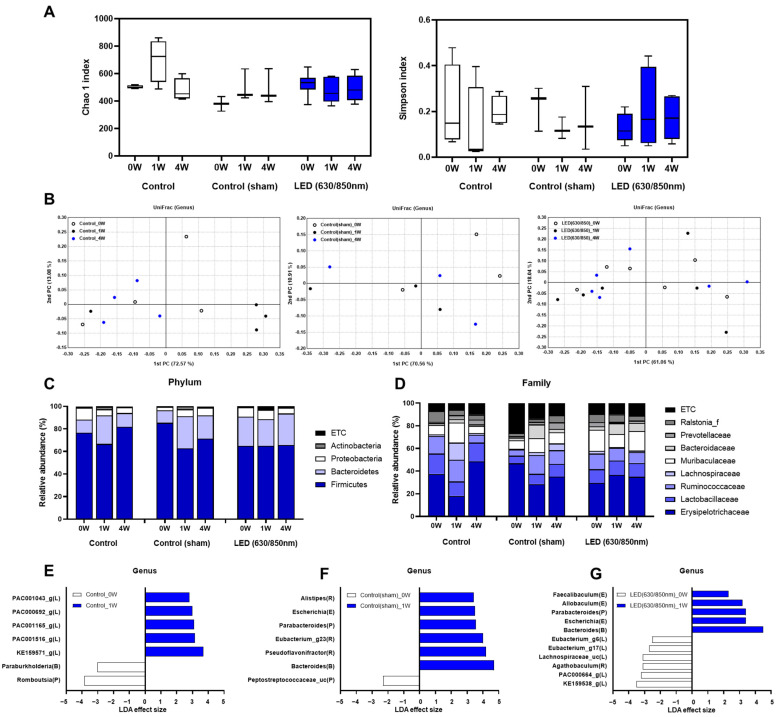
Comparison of gut microbiome after duodenal dual-wavelength photobiomodulation (D-PBM) using micro-LED in Goto–Kakizaki (GK) rats. (**A**) Chao1 index and Simpson index reflecting the richness and diversity of the gut microbiome (no significant difference). (**B**) Principal coordinate analysis (PCoA) plot using Unifrac distance of gut microbial communities obtained from the three groups at different time points. (**C**) Composition profiles of the gut microbiome at the phylum level in control GK, sham, and dual LE ratsD. (**D**) Main fecal microbiomes (family level) in the three groups at different time points. Differentially represented genera between baseline and 1 week in (**E**) control, (**F**) sham control, and (**G**) D-PBM group by linear discriminant analysis (LDA) effect size (LEfSe). Only the bacterial taxa with an LDA score greater than 2 reveal shows statistically significant ranking (*p* < 0.05) of abundant microbiomes on genus level.

**Figure 6 cells-11-03490-f006:**
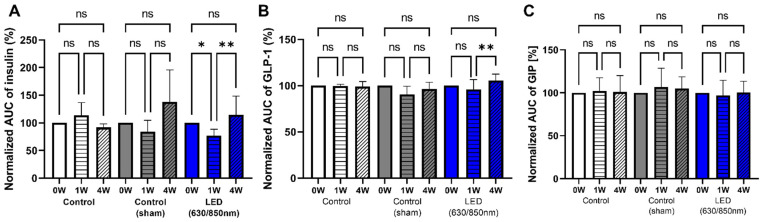
Effects of duodenal dual-wavelength (D) photobiomodulation (PBM) with light-emitting diode (LED) on the production of insulin, glucagon-like peptide 1 (GLP-1), and glucose-dependent insulinotropic polypeptide (GIP) at the baseline and 1 and 4 weeks after the procedure. Serum concentrations of insulin, GLP-1, and GIP were measured using ELISA (Control n = 5, sham n = 5, and dual-wavelength as 630/850 nm, n = 7). Values are expressed as mean ± SEM. Area under the curve (AUC) during a 120-min oral glucose tolerance test at baseline (0 weeks), after 1 week, and after 4 weeks. (**A**) Insulin-AUC was significantly increased in the sham control and duodenal dual-wavelength photobiomodulation (D-PBM) (630/850 nm) groups between 0 versus 1 week (*p* < 0.05) and 1 versus 4 weeks (*p* < 0.01). (**B**) Differences in GLP-1 significantly increased (*p* < 0.01) in the D-PBM group (630/850 nm) between 1 and 4 weeks. (**C**) GIP serum levels showed no differences among the groups. * *p* < 0.05; ** *p* < 0.01.

## Data Availability

Not applicable.

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
