# Peer review of "Duodenal Dual-Wavelength Photobiomodulation Improves Hyperglycemia and Hepatic Parameters with Alteration of Gut Microbiome in Type 2 Diabetes Animal Model"

_cells, 2022, doi:10.3390/cells11213490_

Round 1

Reviewer 1 Report

1. If we are talking about duodenal exposure through a light guide for the purpose of selectivity and availability of light energy, then it is highly desirable to justify this at least theoretically, what exactly (cells, organ) absorbs, how and how much better it absorbs. 2. What was the rationale for choosing the wavelength? Did the authors just take what was at hand? For different wavelengths, absorption can vary by an order of magnitude.

As the authors write, light is scattered externally, it would be better to shine through the endoscope closer to the object (pancreas). But how much light is weakened when exposed externally, due to which, how much is needed outside and through the light guide, for example, externally 100 mW, as authots write (although this is far otherwise - see table below), and only 5 mW from the inside, from where does it follow? + there is also a liver, and a microbiome, how about them? The general idea is this - there is practically no justification for illumination through a light guide. 

Authors / Wavelength, nm / Power, mW

Silva, G., et al., 2000 / 630 nm / 300 mW

Huang, H.-H., et al., 2022 / 633 nm / 15?

Bicknell, B., et al., 2019* / 660 and 808 / 75 and 80 mW

* 250Hz modulation, microbiome study

Reviewer 2 Report

Dear authors, the study is very interesting and brings to the literature a new target in the body to be irradiated and treat diabetes mellitus. However, I have several considerations that need to be clarified:

Abstract

(lines 25,26): the authors stated “Multiple sessions of external photobiomodulation (PBM) in previous animal studies demonstrated improved hyperglycemia, glucose intolerance, and insulin resistance despite the underlying mechanism and target organ being unknown”. However, several animal models showed mechanisms of action of the PBM on insulin resistance and/or blood glucose control. This sentence it is not completely true. Please, change it.

Introduction

(line 72): 75-100 mW is not power density (mW/cm2). Please, revise this information

(line 82): is “19” a reference? If yes, please, add backets.

Material and methods

Animals: Why the authors used Diabetic Goto–Kakizaki (GK) rats? Please, add an explanation about what kind type of diabetes this animal model represents, and why it was chosen. Can this animal model represent both type 1 and type 2 diabetes? Can the results be applied for both types of diabetes? Please, justify.   

Why were the animals housed in cages individually? Could this promote any loss for the animals?

Supplementary material (experiment): the authors reported: “Catheter based radiofrequency ablation (RFA, 2.2cm length, 7W, 5 seconds Taewoong Medical) was applied for simulated duodenal mucosal resurfacing because hydrothermal RFA probe was not commercially available in Korea. Due to the severe burning effect of RFA for duodenum and early death in prior energy settings in SD rats, duration of RFA was limited to 5 seconds”. So, was the e ablation simulated or not? After seeing the supplementary figure 2, the temperature in RFA procedure is 66 °C suggesting a not simulated procedure. Please, add correction or explanation.  

Supplementary material (figure table 1): Why the authors used different prototypes of catheter plus LEDs (triangle and square catheter shapes) for experiment of 1 week and 4 weeks? In addition, please, add the reason and justify according to the literature why the authors applied the double of energy density in 4 weeks experiment compared to 1 week experiment regarding single or dual wavelength? Moreover, why the authors applied the double of energy density comparing single x dual wavelength in 1 week or 4 weeks experiments? How could the dose response affect the results? Was the experiment appropriately designed (regarding the energy density) to respond if dual wavelength is better or not than single wavelength? Finally, why the authors did not test the 850 nm isolated in 4 wks experiment? As demonstrated in figure 1C (not supplementary material), the 850 nm produced better results (approximately 4 times more reduction) than 630 nm regarding AUC of OGTT test. Please, justify!

Supplementary material (table 1): the authors describe in the supplementary text that the 5 cm length of the first part of duodenum was irradiated. And probably used 5 cm as the area to calculate the energy and power density of the photobiomodulation therapy, since the energy applied for example in 630 nm was 10.8 J resulting in 2.16 J/cm2. Was this calculation correct? The photobiomodulation was applied to 5 cm/2 or 5 cm length? For a simple example, note that the triangular catheter is composed by 3 rectangles of 2 mm x 50 mm each. Again, is the area in this catheter equal to 5 cm2? Moreover, seeing the supplementary figure 1, I could perceive that the LEDs are spaced from each other. What is this distance? How were the power of the light (mW) and power density (mW/cm2) measured? What is the power and power density of each LED in red and infrared spectra? Is the numbers presented in table 1 referent to each LED or the prototype? How exactly the power meter was used in the mensuration? The data are definitively not consistent.

Supplementary figure 2: the picture of RFA procedure is completely different from the others regarding PBM. The surgeon’s hand covers almost all body parts of the rat, and maybe affect the analysis of the rat’s body temperature.

Results

The authors stated (line 217-218): “No significant body temperature change was seen in the LED PBM group compared to that in the RFA group. (Figure S2)”. However, the figure 2, specifically the RFA procedure, has a completely different angle of view compared to PBM pictures. In the RFA procedure the surgeon covers almost all body parts of the rat, confounding the rat’s temperature. Please, add new picture with the same results but with the exact angle of view such demonstrated in the pictures of PBM procedure. Otherwise, the result could be biased.

Figure 1C does not present results for RFA procedure as stated in the line 232.

Figure 1D: there is no results (average and standard deviation) in the text (lines 233-234) reporting the total amount of food intake per group.

Figure 1E: the same comments of figure 1D.

There is no report (average and standard deviation) for all measurements at baseline, 1 week and 4 weeks regarding all assessments. This lack of data difficult understanding weight gain or loss (only for simple example) among all groups. I strongly suggest the authors put all data (of all outcomes assessed) in a table as supplementary material for the reviewers and readers.

Figure 3A. definitively is not in the same magnification between groups. Please, revise it.

Figure 5A and B: please, add no significant differences among the groups in the manuscript and figure caption.

Figure 6A and B does not demonstrate significant higher changes between 1 wk and 4 wks in AUC GLP-1 and insulin for sham control, as stated in lines 361-363. Please, revise this information/ results.

Discussion

The authors started the discussion session stating: “This study, using the rodent model of T2DM…”. Again, did the present study use a real T2DM model having an insulin resistance? Or the animals spontaneously had a decreased production of insulin due to a loos of betta cells along the course of life? In addition, make relationship between T2DM in humans and the animal model used in the present study. Both are equal or similar?

The authors stated: No ideal wavelength of LED for PBM has been studied (line 391). So, my question is: What is the ideal wavelength of LED for PBM? The answer should be based on the literature.

All the argue regarding the comparison between single or dual wavelengths applied at the present study, and comparisons with other studies are biased. The reason for this affirmation is that the authors did not take account the total amount of energy or energy density applied between the pilot study and 4wks study, as well as when they compare with previous results published. The author probably knows about the important effect of dose response phenomena in PBM. Sometimes we can see a biphasic or three phasic dose response. Please, reorganize all introduction and discussion to biphasic dose response instead of to dual wavelength or multi-wavelength (that is not suitable, since we have only 2 wavelengths).

I’m agree that duodenum may be a significant therapeutic target for PBM for metabolic diseases including T2DM. And this is an innovation of the present study. In this direction, why the previous studies applied the PBM with higher power of the light (mW) around 75-100 mW (and not power density that is mW/cm2) compared to the power of the light applied in the present study?

The authors stated: “Interestingly, the 4-week follow-up study group with four-dimensional LED had two-fold energy intensity compared to the 1-week pilot study group. However, the reduction of glucose during OGTT in the 4-week follow-up study was modestly increased compared to that in the 1-week pilot study after 1 week of DM-LED PBM (-9.7% in the 1-week pilot study vs. -11.7% in the 4-week follow-up study). This suggests that uniform irradiation with optimal energy intensity in DM-LED PBM may impact the degree of lowering serum glucose levels”. This is the case of dose response. Please, adjust the discussion accordingly.

Regarding the gut microbiome, the authors should be cautions (lines 446-447) in affirm that PBM changes the gut microbiome and due to this change the blood glucose was reduced. The authors did not test it properly. They only checked one of the several effects of the irradiation at duodenum. In addition, the blood glucose control is multifactorial.

The authors stated: “This suggests that dynamic changes in the gut microbiome from 1 week after DM-LED PBM may affect the potentiation of lowering serum glucose levels at 4 weeks (lines 450-451). However, the normalized AUC percentage of insulin decreased significantly from 0 wk to 1 wk in DM-LED group, and after 4 wks it increased significantly (Figure 6A). So, how to explain it? This is not consistent.

The authors stated: “However, our results showed the reduction of liver enzymes such as AST and ALT after DM-LED PBM, suggesting an additional beneficial effect of duodenum LED irradiation on concomitant NAFLD”. I would like to ask the authors if the light applied on the duodenum region could also achieve the liver. Light spread over the duodenum region as showed in supplementary figure 2. Please, discuss it.  

Regarding the last paragraph of discussion, I would like to invite the authors to think about whether the invasive methods of photobiomodulation application is the best option compared to noninvasive. The study brings a very interesting target in the body to be irradiated to treat diabetes. But to make this invasively without any perspective of a definitive cure does not justify it. Probably adjustments in light parameters to irradiate this target tissue through the skin seems to be the best option, mainly thinking in repetitive sessions of photobiomodulation as a noninvasive treatment. Thus, please, give this new direction of discussion instead of adaptations of flexibility of micro LEDs or any similar stuff. 

Reviewer 3 Report

A well designed experiment and a well written paper

Accept as is

Author Response

Thank you for your consideration.

Round 2

Reviewer 2 Report

Dear authors

Thank you for all efforts and changes made in the manuscript. It was improved significantly.

However, the power of the light (W), irradiance (mW/cm2), energy (Joules – J) and fluence or energy density or radiant exposure (J/cm2) remain not answered properly.  Thus, the point 10 and 11 need to be revised.

In the first version of the manuscript (supplementary method – experiment session) the authors reported the use of a power meter Newport 1936-R; Newport Corp., CA, United States. And provided measures described in the table 1 of this supplementary material. However, they did not provide the model of the sensor or light detector used together with the console meter (Newport 1936-R; Newport Corp., CA, United States).

Comparing table 1 (first version) and table 1 (revised version), we can see the same values for power (W), energy (J), irradiance (mW/cm2) and intensity (J/cm2) even using different devices and methods to measure these light parameters (first version versus revised version). It is not possible. My suggestions are:

1- provide the model of the light detector or sensor used together with Newport 1936-R console

2- provide the area (cm2) of the detector or sensor

3- provide the area (cm2) of each mini-LED used to build the catheter prototyped

4- measuring the power of the light (W) and irradiance (mW/cm2): put the central area of the sensor exactly over 1 mini-LED. If the area of the sensor can cover only one mini-LED, divide the power (W) measured by the area of the mini-LED to get the correct irradiance (mW/cm2).  If the area of the sensor is higher than the area of each mini-LED, put the sensor over 2 or 3 mini-LEDs (depending on the area of the sensor) and get the power of the light (W). In this case, the irradiance (mW/cm2) needs to be referred to the area of the sensor and not to the area of each mini-LED, or only the length (cm) of the catheter, as the authors provided in the revised version of the manuscript. Please, correct it. In addition, since the correct power (W) and irradiance (mW/cm2) were measured, and energy (J) calculated properly, the energy density (J/cm2) can be corrected too. Remember that the same area used to find the correct irradiance must be used to find the energy density (J/cm2). Finally, it is necessary correct all these information in the manuscript and back to mW/cm2 and J/cm2 instead of mW/cm or J/cm.

Please, confirm if the square shape catheter had 10 or 20 mini-LEDs. Observing the figure of supplementary material, my thought is that square shape has 20 mini-LEDs (a junction of 2 triangular shape catheters).

Sorry for insist in many questions and corrections. Thanks again.
